# AutoGT: Automated Graph Transformer Architecture Search

**Zizhao Zhang[1], Xin Wang[1,2]\*, Chaoyu Guan[1], Ziwei Zhang[1], Haoyang Li[1], Wenwu Zhu[1]\***

[1]Department of Computer Science and Technology, Tsinghua University
[2]THU-Bosch JCML Center, Tsinghua University
`{zzz22, guancy19, lihy18}@mails.tsinghua.edu.cn`
`{xin_wang, zwzhang, wwzhu}@tsinghua.edu.cn`

## Abstract

Although Transformer architectures have been successfully applied to graph data with the advent of Graph Transformer, the current design of Graph Transformers still heavily relies on human labor and expertise knowledge to decide on proper neural architectures and suitable graph encoding strategies at each Transformer layer. In literature, there have been some works on the automated design of Transformers focusing on non-graph data such as texts and images without considering graph encoding strategies, which fail to handle the non-euclidean graph data. In this paper, we study the problem of automated graph Transformers, for the first time. However, solving these problems poses the following challenges: i) how can we design a unified search space for graph Transformer, and ii) how to deal with the coupling relations between Transformer architectures and the graph encodings of each Transformer layer. To address these challenges, we propose Automated Graph Transformer (AutoGT), a neural architecture search framework that can automatically discover the optimal graph Transformer architectures by joint optimization of Transformer architecture and graph encoding strategies. Specifically, we first propose a unified graph Transformer formulation that can represent most state-of-the-art graph Transformer architectures. Based upon the unified formulation, we further design the graph Transformer search space that includes both candidate architectures and various graph encodings. To handle the coupling relations, we propose a novel encoding-aware performance estimation strategy by gradually training and splitting the supernets according to the correlations between graph encodings and architectures. The proposed strategy can provide a more consistent and fine-grained performance prediction when evaluating the jointly optimized graph encodings and architectures. Extensive experiments and ablation studies show that our proposed AutoGT gains sufficient improvement over state-of-the-art hand-crafted baselines on all datasets, demonstrating its effectiveness and wide applicability.

## 1 Introduction

Recently, designing Transformer for graph data has attracted intensive research interests (Dwivedi & Bresson, 2020; Ying et al., 2021). As a powerful architecture to extract meaningful information from relational data, the graph Transformers have been successfully applied in natural language processing (Zhang & Zhang, 2020; Cai & Lam, 2020; Wang et al., 2023), social networks (Hu et al., 2020b), chemistry (Chen et al., 2019; Rong et al., 2020), recommendation (Xia et al., 2021) etc. However, developing a state-of-the-art graph Transformer for downstream tasks is still challenging because it heavily relies on the tedious trial-and-error hand-crafted human design, including determining the best Transformer architecture and the choices of proper graph encoding strategies to utilize, etc. In addition, the inefficient hand-crafted design will also inevitably introduce human bias, which leads to sub-optimal solutions for developing graph transformers. In literature, there have been works on automatically searching for the architectures of Transformer, which are designed specifically for data

---

*Corresponding authors

in Natural Language Processing (Xu et al., 2021) and Computer Vision (Chen et al., 2021b). These works only focus on non-graph data without considering the graph encoding strategies which are shown to be very important in capturing graph information (Min et al., 2022a), thus failing to handle graph data with non-euclidean properties.

In this paper, we study the problem of automated graph Transformers for the first time. However, previous work (Min et al., 2022a) has demonstrated that a good graph Transformer architecture is expected to not only select proper neural architectures for every layer but also utilize appropriate encoding strategies capable of capturing various meaningful graph structure information to boost graph Transformer performance. Therefore, there exist two critical challenges for automated graph Transformers:

- **How to design a unified search space appropriate for graph Transformer?** A good graph Transformer needs to handle the non-euclidean graph data, requiring explicit consideration of node relations within the search space, where the architectures, as well as the encoding strategies, can be incorporated simultaneously.
- **How to conduct encoding-aware architecture search strategy to tackle the coupling relations between Transformer architectures and graph encoding?** Although one simple solution may resort to a one-shot formulation enabling efficient searching in vanilla Transformer operation space which can change its functionality during supernet training, the graph encoding strategies differ from vanilla Transformer in containing certain meanings related to structure information. How to train an encoding-aware supernet specifically designed for graphs is challenging.

To address these challenges, we propose Automated Graph Transformer, AutoGT[1], a novel neural architecture search method for graph Transformer. In particular, we propose a unified graph Transformer formulation to cover most of the state-of-the-art graph Transformer architectures in our search space. Besides the general search space of the Transformer with hidden dimension, feed-forward dimension, number of attention head, attention head dimension, and number of layers, our unified search space introduces two new kinds of augmentation strategies to attain graph information: node attribution augmentation and attention map augmentation. To handle the coupling relations, we further propose a novel encoding-aware performance estimation strategy tailored for graphs. As the encoding strategy and architecture have strong coupling relations when generating results, our AutoGT split the supernet based on the important encoding strategy during evaluation to handle the coupling relations. As such, we propose to gradually train and split the supernets according to the most coupled augmentation, attention map augmentation, using various supernets to evaluate different architectures in our unified searching space, which can provide a more consistent and fine-grained performance prediction when evaluating the jointly optimized architecture and encoding. In summary, we made the following contributions:

- We propose Automated Graph Transformer, AutoGT, a novel neural architecture search framework for graph Transformer, which can automatically discover the optimal graph Transformer architectures for various down-streaming tasks. To the best of our knowledge, AutoGT is the first automated graph Transformer framework.
- We design a unified search space containing both the Transformer architectures and the essential graph encoding strategies, covering most of the state-of-the-art graph Transformer, which can lead to global optimal for structure information excavation and node information retrieval.
- We propose an encoding-aware performance estimation strategy tailored for graphs to provide a more accurate and consistent performance prediction without bringing heavier computation costs. The encoding strategy and the Transformer architecture are jointly optimized to discover the best graph Transformers.
- The extensive experiments show that our proposed AutoGT model can significantly outperform the state-of-the-art baselines on graph classification tasks over several datasets with different scales.

## 2 RELATED WORK

**The Graph Transformer.** Graph Transformer, as a category of neural networks, enables Transformer to handle graph data (Min et al., 2022a). Several works (Dwivedi & Bresson, 2020; Ying et al.,

---

[1]Our codes are publicly available at https://github.com/SandMartex/AutoGT

2021; Hussain et al., 2021; Zhang et al., 2020; Kreuzer et al., 2021; Shi et al., 2021) propose to pre-calculate some node positional encoding from graph structure and add them to the node attributes after a linear or embedding layer. Some works (Dwivedi & Bresson, 2020; Zhao et al., 2021; Ying et al., 2021; Khoo et al., 2020) also propose to add manually designed graph structural information into the attention matrix in Transformer layers. Others (Yao et al., 2020; Min et al., 2022b) explore the mask mechanism in the attention matrix, masking the influence of non-neighbor nodes. In particular, UniMP (Shi et al., 2021) achieves new state-of-the-art results on OGB (Hu et al., 2020a) datasets, Graphormer (Ying et al., 2021) won first place in KDD Cup Challenge on Large-SCale graph classification by encoding various information about graph structures into graph Transformer.

**Neural Architecture Search.** Neural architecture search has drawn increasing attention in the past few years (Elsken et al., 2019; Zoph & Le, 2017; Ma et al., 2018; Pham et al., 2018; Wei et al., 2021; Cai et al., 2022; Guan et al.; 2021b; 2022; Qin et al., 2022a;b; Zhang et al., 2021; **?**). There are many efforts to automate the design of Transformers. (So et al., 2019) propose the first automated framework for Transformer in neural machine translation tasks. AutoTrans (Zhu et al., 2021) improves the search efficiency of the NLP Transformer through a one-shot supernet training. NAS-BERT (Xu et al., 2021) further leverages the neural architecture search for big language model distillation and compression. AutoFormer (Chen et al., 2021b) migrates the automation of the Transformer for vision tasks, where they utilize weight-entanglement to improve the consistency of the supernet training. GLiT (Chen et al., 2021a) proposes to search both global and local attention for the Vision Transformer using a hierarchical evolutionary search algorithm. (Chen et al., 2021c) further propose to evolve the search space of the Vision Transformer to solve the exponential explosion problems.

## 3 AUTOMATED GRAPH TRANSFORMER ARCHITECTURE SEARCH (AUTOGT)

To automatically design graph Transformer architectures, we first unify the formulation of current graph Transformers in Section 3.1. Based on the unified formulation, we design the search space tailored for the graph Transformers in Section 3.2. We propose a novel encoding-aware performance estimation strategy in Section 3.3, and introduce our evolutionary search strategy in Section 3.4. The whole algorithm is presented by Figure 2.

### 3.1 THE UNIFIED GRAPH TRANSFORMER FRAMEWORK

Current representative graph Transformer designs can be regarded as improving the input and attention map in Transformer architecture through various graph encoding strategies. We first introduce the basic Transformer architecture and then show how to combine various graph encoding strategies.

Let $G = (V, E)$ denote a graph where $V = \{v_1, v_2, \cdots, v_n\}$ represents the set of nodes and $E = \{e_1, e_2, \cdots, e_m\}$ represents the set of edges, and denote $n = |V|$ and $m = |E|$ as the number of nodes and edges, respectively. Let $\mathbf{v}_i, i \in \{1, ..., n\}$ represents the features of node $v_i$, and $\mathbf{e}_j, j \in \{1, ..., m\}$ represents the features of edge $e_j$.

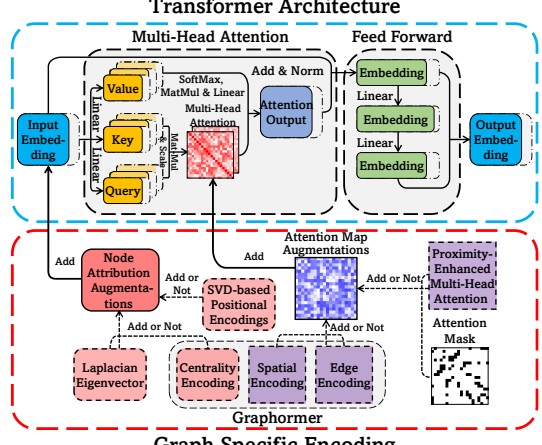

Figure 1: The unified graph Transformer search space. It consists of the Transformer architecture space and the graph specific encoding space. The Transformer architecture search space is detailed in Table 1. The graph specific encoding search space is to decide whether each encoding strategy should be adopted or not and the mask threshold for the attention mask.

#### 3.1.1 BASIC TRANSFORMER

As shown in Figure 1, a basic Transformer consists of several stacked blocks, with each block containing two modules, namely the multi-head attention (MHA) module and the feed-forward network (FFN) module.

At block $l$, the node representation $\mathbf{H}^{(l)} \in \mathbb{R}^{n \times d}$ first goes through the MHA module to interact with each other and pass information through self-attention:

$$\mathbf{A}_h^{(l)} = \mathrm{softmax}\left(\frac{\mathbf{Q}_h^{(l)}\mathbf{K}_h^{(l)T}}{\sqrt{d_k}}\right), \mathbf{O}_h^{(l)} = \mathbf{A}_h^{(l)}\mathbf{V}_h^{(l)}, \tag{1}$$

where $\mathbf{A}_h^{(l)} \in \mathbb{R}^{n \times n}$ is the message passing matrix, $\mathbf{O}_h^{(l)}$ is the output of the self-attention mechanism of the $h^{th}$ attention head, $h = 1, 2, \cdots, Head$, $Head$ is the number of attention heads, and $\mathbf{K}_h^{(l)}, \mathbf{Q}_h^{(l)}, \mathbf{V}_h^{(l)} \in \mathbb{R}^{n \times d_k}$ are the key, query, value calculated as:

$$\mathbf{K}_h^{(l)}, = \mathbf{H}^{(l)}\mathbf{W}_{k,h}^{(l)}, \mathbf{Q}_h^{(l)} = \mathbf{H}^{(l)}\mathbf{W}_{q,h}^{(l)}, \mathbf{V}_h^{(l)} = \mathbf{H}^{(l)}\mathbf{W}_{v,h}^{(l)}, \tag{2}$$

where $\mathbf{W}_{k,h}^{(l)}, \mathbf{W}_{q,h}^{(l)}, \mathbf{W}_{v,h}^{(l)} \in \mathbb{R}^{d \times d_k}$ are learnable parameters. Then, the representations of different heads are concatenated and further transformed as:

$$\mathbf{O}^{(l)} = (\mathbf{O}_1^{(l)} \circ \mathbf{O}_2^{(l)} \circ ... \circ \mathbf{O}_{Head}^{(l)})\mathbf{W}_O^{(l)} + \mathbf{H}^{(l)}, \tag{3}$$

where $\mathbf{W}_O^{(l)} \in \mathbb{R}^{(d_k * Head) \times d_t}$ is the parameter and $\mathbf{O}^{(l)}$ is the multi-head result. Then, the attended representation will go through the FFN module to further refine the information of each node:

$$\mathbf{H}^{(l+1)} = \sigma(\mathbf{O}^{(l)}\mathbf{W}_1^{(l)})\mathbf{W}_2^{(l)}, \tag{4}$$

where $\mathbf{O}^{(l)} \in \mathbb{R}^{n \times d_t}$ is the output, $\mathbf{W}_1^{(l)} \in \mathbb{R}^{d_k \times d_h}$, $\mathbf{W}_2^{(l)} \in \mathbb{R}^{d_h \times d}$ are weight matrices.

As for the input of the first block, we concatenate all the node features $\mathbf{H}^{(0)} = [\mathbf{v}_1, ..., \mathbf{v}_n]$. After $L$ blocks, we obtain the final representation of each node $\mathbf{H}^{(L)}$.

### 3.1.2 GRAPH ENCODING STRATEGY

From Section 3.1.1, we can observe that directly using the basic Transformer on graphs can only process node attributes, ignoring important edge attributes and graph topology information in the graph. To make the Transformer architecture aware of the graph structure, several works resort to various graph encoding strategies, which can be divided into two kinds of categories: node attribution augmentation and attention map augmentation.

The node attribution augmentations take the whole graph $G$ as input and generate the topology-aware features $Enc_{node}(G)$ for each node to directly improve the node representations:

$$\mathbf{H}_{aug}^{(l)} = \mathbf{H}^{(l)} + Enc_{node}(G). \tag{5}$$

On the other hand, the attention map augmentations generate an additional attention map $Enc_{map}(G)$, which represents the relationships of any two nodes and improves the attention map generated by self-attention in Eq equation 1 as:

$$\mathbf{A}_{h,aug}^{(l)} = \mathrm{softmax}\left(\frac{\mathbf{Q}_h^{(l)}\mathbf{K}_h^{(l)T}}{\sqrt{d}} + Enc_{map}(G)\right). \tag{6}$$

Combining node attribution augmentations and attention map augmentations together, our proposed framework is as follows:

$$\mathbf{H}^{(l+1)} = \sigma\left(\mathrm{Concat}\left(\mathrm{softmax}\left(\frac{\mathbf{H}_{aug}^{(l)}\mathbf{W}_{q,h}^{(l)}(\mathbf{H}_{aug}^{(l)}\mathbf{W}_{k,h}^{(l)})^T}{\sqrt{d_k}} + Enc_{map}(G)\right)\mathbf{H}_{aug}^{(l)}\mathbf{W}_{v,h}^{(l)}\right)\mathbf{W}_1^{(l)}\right)\mathbf{W}_2^{(l)}. \tag{7}$$

where $\mathbf{H}_{aug}^{(l)} = \mathbf{H}^{(l)} + Enc_{node}(G)$.

### 3.2 THE GRAPH TRANSFORMER SEARCH SPACE

Based on the unified graph Transformer formulation, we propose our unified search space design, which can be decomposed into two parts, i.e., Transformer Architecture space and graph encoding space. Figure 1 shows the unified graph Transformer search space.

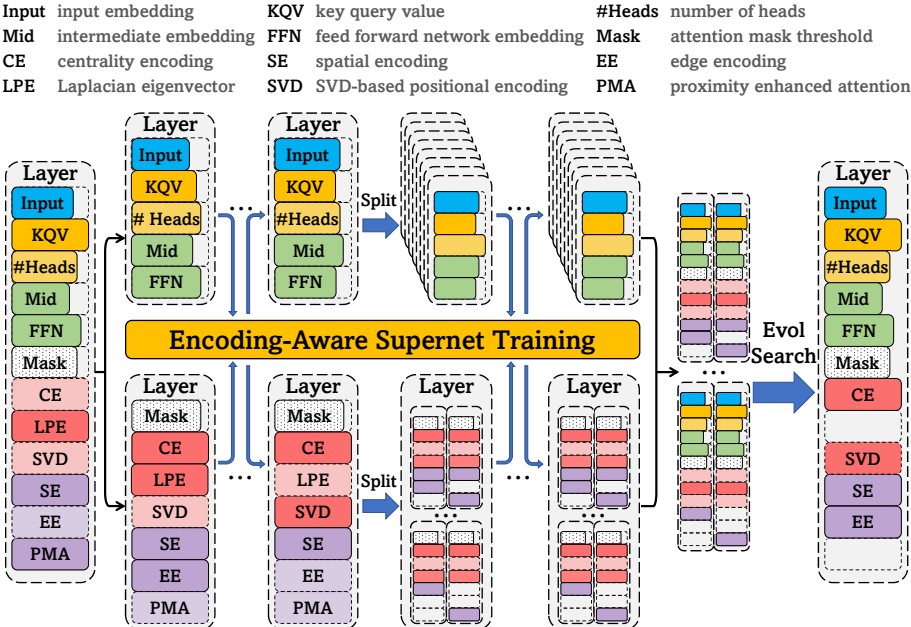

Figure 2: The framework of our work. Firstly, we construct the search space for each layer, consisting of the Transformer architecture space (above) and the graph encoding strategy space (below). Then, we carry out our encoding-aware supernet training method in two stages: before splitting, we train a supernet by randomly sampling architectures from the search space, while after splitting, we train multiple subnets (inheriting the weights from the supernet) by randomly sampling architectures with fixed attention map augmentation strategies (except for the attention mask). Finally, we conduct an evolutionary search based on the subnets and obtain our final architecture and results.

### 3.2.1 TRANSFORMER ARCHITECTURE SPACE

Following Section 3.1.1, we automate five key architecture components for graph Transformer as follows: the number of encoder layers $L$, the dimension $d$, the intermediate dimension $d_t$, the hidden dimension $d_h$, the number of attention heads $Head$, and the attention head dimension $d_k$ in graph Transformer. Notice that these five components already cover the most important designs for Transformer architectures.

A suitable search space should be not only expressive enough to allow powerful architectures, but also compact enough to enable efficient searches. With this principle in mind, we propose two search spaces for these components with different size ranges. Table 1 gives the detailed search space for these two spaces.

### 3.2.2 GRAPH ENCODING SPACE

To exploit the potential of the graph encoding strategies, we further determine whether and which graph encoding strategies to use for each layer of the graph Transformer. Specifically, we explore the node attribution augmentations encoding and attention map augmentations encoding as below.

**Node Attribution Augmentations:**

- **Centrality Encoding** (Ying et al., 2021). Use two node embeddings with the same size representing the in-degree and the out-degree of nodes, i.e.,

$$h_i^{(l)} = x_i^{(l)} + z_{deg^-(v_i)}^- + z_{deg^-(v_i)}^+ \tag{8}$$

  where $h_i^{(l)}$ is the input embedding in layer $l$, $x_i$ is the input attribution of node $i$ in layer $l$, and $z^-$ and $z^+$ are the embedding generated by the in-degree and out-degree.

- **Laplacian Eigenvector** (Dwivedi & Bresson, 2020). Conducting spectral decomposition of the graph Laplacian matrix:

$$\mathbf{U}^T \Lambda \mathbf{U} = \mathbf{I} - \mathbf{D}^{-1/2} \mathbf{A}^G \mathbf{D}^{-1/2} \tag{9}$$

Table 1: The Transformer Architecture Search Space for $\text{AutoGT}_{\text{base}}$ and $\text{AutoGT}$.

| | $\text{AutoGT}_{\text{base}}$ | | $\text{AutoGT}$ | |
| --- | --- | --- | --- | --- |
| | Choices | Supernet Size | Choices | Supernet Size |
| #Layers | {2,3,4} | 4 | {5,6,7,8} | 8 |
| Input Dimension $d$ | {24,28,32} | 32 | {96,112,128} | 128 |
| Intermediate Dimension $d_t$ | {24,28,32} | 32 | {96,112,128} | 128 |
| Hidden Dimension $d_h$ | {24,28,32} | 32 | {96,112,128} | 128 |
| #Attention Heads | {2,3,4} | 4 | {6,7,8} | 8 |
| Attention Head Dimension $d_k$ | {6,8} | 8 | {12,14,16} | 16 |

where $\mathbf{A}^G$ is the adjacency matrix of graph $G$, $\mathbf{D}$ is the diagonal degree matrix, and $\mathbf{U}$ and $\Lambda$ are the eigenvectors and eigenvalues, respectively. We only select the eigenvectors of the $k$ smallest non-zero eigenvalues as the final embedding and concatenate them to the input node attribute matrix for each layer.

- **SVD-based Positional Encoding** (Hussain et al., 2021). Conducting singular value decomposition to the graph adjacency matrix:

$$\mathbf{A}^G \overset{\text{SVD}}{\approx} \mathbf{U}\mathbf{\Sigma}\mathbf{V}^T = (\mathbf{U}\sqrt{\mathbf{\Sigma}}) \cdot (\mathbf{V}\sqrt{\mathbf{\Sigma}})^T = \hat{\mathbf{U}}\hat{\mathbf{V}}^T \tag{10}$$

where $\mathbf{U}, \mathbf{V} \in \mathbb{R}^{n \times r}$ contains the left and right singular vectors of the top $r$ singular values in the diagonal matrix $\mathbf{\Sigma} \in \mathbb{R}^{r \times r}$. Without loss of generality, we only choose $\hat{\mathbf{U}}$ as final embedding since they are highly correlated for symmetric graphs (with differences in signs, to be specific). Similar to Laplacian eigenvector, we concatenate it to input node attribute matrix for each layer.

**Attention Map Augmentations Space:**

- **Spatial Encoding** (Ying et al., 2021). Spatial encoding is added to the attention result before softmax:

$$A_{ij} = \frac{(h_i W_Q)(h_j W_K)^T}{\sqrt{d_k}} + b_{\phi(v_i, v_j)} \tag{11}$$

where $\phi(v_i, v_j)$ is the length of the shortest path from $v_i$ to $v_j$, and $b \in \mathbb{R}$ is a weight parameter generated by $\phi(v_i, v_j)$.

- **Edge Encoding** (Ying et al., 2021). Edge encoding is added to the attention result before softmax:

$$A_{ij} = \frac{(h_i W_Q)(h_j W_K)^T}{\sqrt{d_k}} + \frac{1}{N} \sum_{n=1}^{N} x_{e_n} (w_n^E)^T \tag{12}$$

where $x_{e_n}$ is the feature of the $n$-th edge $e_n$ on the shortest path between $v_i$ and $v_j$, and $w_n^E$ is the $n$-th learnable embedding vector.

- **Proximity-Enhanced Attention** (Zhao et al., 2021). Proximity-Enhanced Attention is added to the attention result before softmax:

$$A_{ij} = \frac{(h_i W_Q)(h_j W_K)^T}{\sqrt{d_k}} + \phi_{ij}^T b \tag{13}$$

where $b \in \mathbb{R}^{M \times 1}$ is a learnable parameter, $\phi_{ij} = \text{Concat}(\Phi_m(v_i, v_j) | m \in 0, 1, \cdots, M-1)$ is the structural encoding generated from: $\Phi_m(v_i, v_j) = \tilde{\mathbf{A}}^m[i, j]$, where $\tilde{\mathbf{A}} = \text{Norm}(\mathbf{A} + \mathbf{I})$ represents the normalized adjacency matrix. Thus the augmentation denotes the reachable probabilities between nodes.

- **Attention Mask** (Min et al., 2022b; Yao et al., 2020). Attention Mask is added to the attention result before softmax:

$$A_{ij} = \frac{(h_i W_Q)(h_j W_K)^T}{\sqrt{d_k}} + \text{Mask}_m(v_i, v_j) \tag{14}$$

where $m$ is the mask threshold, $\text{Mask}_m(v_i, v_j)$ depends on the relationship between $m$ and $\phi(v_i, v_j)$, i.e. the shortest path length between $v_i$ and $v_j$. When $m \geq \phi(v_i, v_j)$, $\text{Mask}_m(v_i, v_j) = 0$. Otherwise, $\text{Mask}_m(v_i, v_j)$ is $-\infty$, masking the corresponding attention in practical terms.

Table 2: Comparison of our proposed unified framework with state-of-the-art graph Transformer models. CE, LPE, SVD, SE, EE, PMA, Mask denote Centrality Encoding, Laplacian Eigenvector, SVD-based Positional Encoding, Spatial Encoding, Edge Encoding, Proximity-Enhanced Attention, and Attention Mask respectively.

| | CE | LPE | SVD | SE | EE | PMA | Mask |
|---|---|---|---|---|---|---|---|
| EGT (Hussain et al., 2021) | | | ✓ | | | | |
| Gophormer (Zhao et al., 2021) | | | | | | ✓ | |
| Graph Trans (Dwivedi & Bresson, 2020) | | ✓ | | | | | ✓ |
| Graphormer (Ying et al., 2021) | ✓ | | | ✓ | ✓ | | |
| Ours | ✓ | ✓ | ✓ | ✓ | ✓ | ✓ | ✓ |

### 3.3 ENCODING-AWARE SUPERNET TRAINING

We next introduce our proposed encoding-aware performance estimation strategy for efficient training.

Similar to general NAS problems, the graph Transformer architecture search can be formulated as a bi-level optimization problem:

$$a^* = \text{argmax}_{a \in \mathcal{A}} Acc_{val}(\mathbf{W}^*(a), a), \qquad \text{s.t.} \mathbf{W}^*(a) = \text{argmin}_{\mathbf{W}} \mathcal{L}_{train}(\mathbf{W}, a), \qquad (15)$$

where $a \in \mathcal{A}$ is the architecture in the search space $\mathcal{A}$, $Acc_{val}$ stands for the validation accuracy, $\mathbf{W}$ represents the learnable weights, and $a^*$ and $\mathbf{W}^*(a)$ denotes the optimal architecture and the optimal weights for the architecture $a$.

Following one-shot NAS methods (Liu et al., 2019; Pham et al., 2018), we encode all candidate architectures in the search space into a supernet and transform Eq. equation 15 into a two-step optimization (Guo et al., 2020):

$$a^* = \text{argmax}_{a \in \mathcal{A}} Acc_{val}(\mathbf{W}^*, a), \qquad \mathbf{W}^* = \text{argmin}_{\mathbf{W}} \mathbb{E}_{a \in \mathcal{A}} \mathcal{L}_{train}(\mathbf{W}, a), \qquad (16)$$

where $\mathbf{W}$ denotes the shared learnable weights in the supernet with its optimal value $\mathbf{W}^*$ for all the architectures in the search space.

To further improve the optimization efficiency of the supernet training, we leverage weight entanglement (Guan et al., 2021a; Chen et al., 2021b; Guo et al., 2020) to deeply share the weights of architectures with different hidden sizes. Specifically, for every architecture sampled from the supernet, we use a 0-1 mask to discard unnecessary hidden channels instead of maintaining a new set of weights. In this way, the number of parameters in the supernet will remain the same as the largest (i.e., with the most parameters) model in the search space, thus leading to efficient optimization.

Although this strategy is fast and convenient, using the same supernet parameters $\mathbf{W}$ for all architectures will decrease the consistency between the estimation of the supernet and the ground-truth architecture performance. To improve the consistency and accuracy of supernet, we propose an encoding-aware supernet training strategy. Based on the contribution of coupling of different encoding strategies, we split the search space into different sub-spaces based on whether adopting three kinds of attention map augmentation strategies: spatial encoding, edge encoding, and proximity-enhanced attention. Therefore, there are $2^3 = 8$ supernets.

To be specific, we first train a single supernet for certain epochs and split the supernet into 8 subnets according to the sub-spaces afterward. Then, we continuously train the weights in each subnet $\mathbf{W}_i$ by only sampling the architecture from the corresponding subspace $A_i$. Experiments to support such a design are provided in Section 4.

### 3.4 EVOLUTIONARY SEARCH

Similar to other NAS research, our proposed graph transformer search space is too large to enumerate. Therefore, we propose to utilize the evolutionary algorithm to efficiently explore the search space to obtain the architecture with optimal accuracy on the validation dataset.

Specifically, we first maintain a population consisting of $T$ architectures by random sample. Then, we evolve the architectures through our designed mutation and crossover operations. In the mutation operation, we randomly choose from the top-$k$ architectures with the highest performance in the

Table 3: Comparisons of AutoGT against state-of-the-art hand-crafted baselines. We report the average accuracy (%) and the standard deviation on all the datasets. Out-of-time (OOT) indicates the method cannot produce results in 1 GPU day.

| Dataset | COX2_MD | BZR_MD | PTC_FM | DHFR_MD | PROTEINS | DBLP |
|---|---|---|---|---|---|---|
| GIN | $45.82_{14.35}$ | $59.68_{14.65}$ | $57.87_{8.86}$ | $62.88_{8.26}$ | $73.76_{4.61}$ | $91.18_{0.42}$ |
| DGCNN | $54.81_{18.51}$ | $62.74_{20.59}$ | $62.17_{3.62}$ | $63.89_{5.91}$ | $72.68_{3.75}$ | $91.57_{0.54}$ |
| DiffPool | $51.45_{14.28}$ | $65.01_{14.74}$ | $60.16_{5.87}$ | $61.06_{9.42}$ | $73.31_{3.75}$ | OOT |
| GraphSAGE | $49.59_{12.80}$ | $57.43_{13.50}$ | $64.17_{3.28}$ | $66.92_{2.35}$ | $67.19_{6.97}$ | $51.01_{0.02}$ |
| Graphormer | $56.39_{15.03}$ | $63.94_{12.58}$ | $64.88_{7.58}$ | $64.88_{7.58}$ | $75.29_{3.10}$ | $89.36_{2.31}$ |
| GT(ours) | $54.44_{16.84}$ | $63.33_{11.67}$ | $64.18_{2.60}$ | $65.68_{5.64}$ | $73.94_{3.78}$ | $90.67_{1.01}$ |
| AutoGT(ours) | $\mathbf{59.72}_{23.26}$ | $\mathbf{65.92}_{10.00}$ | $\mathbf{65.60}_{3.71}$ | $\mathbf{68.22}_{5.02}$ | $\mathbf{77.17}_{3.40}$ | $\mathbf{91.66}_{0.79}$ |

Table 4: Comparisons of AutoGT against state-of-the-art hand-crafted baselines. We report the area under the curve (AUC) [%] and the standard deviation on all the datasets.

| Dataset | OGBG-MolHIV | OGBG-MolBACE | OGBG-MolBBBP |
|---|---|---|---|
| GIN | $71.11_{2.57}$ | $70.42_{4.78}$ | $63.37_{1.81}$ |
| DGCNN | $69.97_{2.16}$ | $75.62_{2.64}$ | $60.92_{1.78}$ |
| DiffPool | $74.58_{1.71}$ | $73.87_{4.50}$ | $66.68_{6.08}$ |
| GraphSAGE | $67.82_{3.67}$ | $72.91_{1.24}$ | $64.19_{3.50}$ |
| Graphormer | $71.89_{2.66}$ | $76.42_{1.67}$ | $66.52_{0.74}$ |
| AutoGT(ours) | $\mathbf{74.95}_{1.02}$ | $\mathbf{76.70}_{1.42}$ | $\mathbf{67.29}_{1.46}$ |

last generation and change its architecture choices with probabilities. In the crossover operation, we randomly select pairs of architectures with the same number of layers from the remaining architectures, and randomly switch their architecture choices.

## 4 EXPERIMENTS

In this section, we present detailed experimental results as well as the ablation studies to empirically show the effectiveness of our proposed AutoGT.

**Datasets and Baselines.** We first consider six graph classification datasets from Deep Graph Kernels Benchmark((Yanardag & Vishwanathan, 2015)) and TUDataset (Morris et al., 2020), namely COX2_MD, BZR_MD, PTC_FM, DHFR_MD, PROTEINS, and DBLP. We also adopt three datasets from Open Graph Benchmark (OGB) (Hu et al., 2020a), including OGBG-MolHIV, OGBG-MolBACE, and OGBG-MolBBBP. The task is to predict the label of each graph using node/edge attributes and graph structures. The detailed statistics of the datasets are shown in Table 6 in the appendix.

We compare AutoGT with state-of-the-art hand-crafted baselines, including GIN (Xu et al., 2019), DGCNN (Zhang et al., 2018), DiffPool (Ying et al., 2018), GraphSAGE (Hamilton et al., 2017), and Graphormer (Ying et al., 2021). Notice that Graphormer is a state-of-the-art graph Transformer architecture that won first place in the graph classification task of KDD Cup 2021 (OGB-LSC).

For all the datasets, we follow Errica et al., (Errica et al., 2020) to utilize 10-fold cross-validation for all the baselines and our proposed method. All the hyper-parameters and training strategies of baselines are implemented according to the publicly available codes (Errica et al., 2020)[2].

**Implementation Details.** Recall that our proposed architecture space has two variants, a larger $\text{AutoGT}(L = 8, d = 128)$ and a smaller $\text{AutoGT}_{base}(L = 4, d = 32)$. In our experiments, we adopt the smaller search space for five relatively small datasets, i.e., all datasets except DBLP, and the larger search space for DBLP. We use the Adam optimizer, and the learning rate is $3e - 4$. For the smaller/larger datasets, we set the number of iterations to split (i.e., $T_s$ in Algorithm 1 in Appendix) as 50/6 and the maximum number of iterations (i.e., $T_m$ in Algorithm 1) as 200/50. The batch size is 128. The hyperparameters of these baselines are kept consistent with our method for a fair comparison.

---

[2]https://github.com/diningphil/gnn-comparison

We also report the results of our unified framework in Section 3.1, i.e. mixing all the encodings in our search space with the supernet but without the search part, denoted as GT(Graph Transformer).

**Experimental Results.** We report the results in Table 3. We can make the following observations. First, AutoGT consistently outperforms all the existing hand-crafted methods on all datasets, demonstrating the effectiveness of our proposed method. Graphormer shows remarkable performance and achieves the second-best results on three datasets, showing the great potential of Transformer architectures in processing graph data. However, since Graphormer is a manually designed architecture and cannot adapt to different datasets, it fails to be as effective as our proposed automatic solution. Lastly, GT, our proposed unified framework, fails to show strong performance in most cases. The results indicate that simply mixing different graph Transformers cannot produce satisfactory results, demonstrating the importance of searching for effective architectures to handle different datasets.

We also conduct experiments on Open Graph Benchmark (OGB) (Hu et al., 2020a). On the three binary classification datasets of OGB, we report the AUC score of our method and all the baselines. The results also show that our method outperforms all the hand-crafted baselines on these datasets.

**Time Cost.** We further show the time comparison of AutoGT with hand-crafted graph transformer Graphormer. On OGBG-MolHIV dataset, both Graphormer and AutoGT cost 2 minutes for one epoch on single GPU. The default Graphormer is trained for 300 epochs, which costs 10 hours to obtain the result of one random seed. For AutoGT, we train shared supernet for 50 epochs, 8 supernets inherit, and continue to train for 150 epochs. So the training process costs totally 1250 epochs with 40 hours. And on the evolutionary search stage, we evaluate 2000 architectures' inheriting weight performances, which costs about 900 epochs with 30 hours. In summary, the total time cost for AutoGT is only 7 times total time cost for a hand-crafted graph transformer Graphormer.

**Ablation Studies.** We verify the effectiveness of the proposed encoding-aware supernet training strategy by reporting the results on the PROTEINS dataset, while other datasets show similar patterns.

To show the importance of considering encoding strategies when training the supernet, we design two variants of AutoGT and compare the results:

- **One-Shot.** We only train a single supernet and use it to evaluate all the architectures.
- **Positional-Aware.** We also split up the supernet into 8 subnets but based on three node attribute augmentations instead of the three attention map augmentation as in AutoGT.

The results of AutoGT and two variants are shown in Table 5. From the table, we can observe that, compared with the result of one-shot NAS, positional-aware and AutoGT methods achieve different levels of improvement. Further comparing the accuracy gain, we find that the result of AutoGT (1.25%) is nearly 5 times larger than the result of positional-aware (0.27%), even though both methods adopt 8 subnets. We attribute the significant difference in accuracy gain from supernet splitting to the different degrees of coupling of graph encoding strategies with the Transformer architecture. For example, the dimensionality of node attribution augmentation is the same as the number of nodes, while the attention map augmentation has a quadratic dimensionality, resulting in different coupling degrees. Our proposed encoding-aware performance estimation based on three attention map augmentation strategies is shown to be effective in practice.

Table 5: The ablation study on the effectiveness of the proposed encoding-aware supernet training strategy. We report the average accuracy[%] with the variance on PROTEINS.

| Method | Accuracy |
|---|---|
| **One-Shot** | $75.92_{3.10}$ |
| **Positional-Aware** | $76.19_{3.42}$ |
| **AutoGT** | $77.17_{3.40}$ |

## 5 CONCLUSION

In this paper, we propose AutoGT, a neural architecture search framework for graph Transformers. We design a search space tailored for graph Transformer architectures, and an encoding-aware supernet training strategy to provide reliable graph Transformer supernets considering various graph encoding strategies. Our method integrates the existing graph Transformer into a unified framework, where different Transformer encodings can enhance each other. Extensive experiments on six datasets demonstrate that our proposed AutoGT consistently outperforms state-of-the-art baselines on all datasets, demonstrating its strength on various graph tasks.

## ACKNOWLEDGEMENTS

This work was supported in part by the National Key Research and Development Program of China No. 2020AAA0106300, National Natural Science Foundation of China (No. 62250008, 62222209, 62102222, 61936011, 62206149), China National Postdoctoral Program for Innovative Talents No. BX20220185, and China Postdoctoral Science Foundation No. 2022M711813, Tsinghua GuoQiang Research Center Grant 2020GQG1014 and partially funded by THU-Bosch JCML Center. All opinions, findings, and conclusions in this paper are those of the authors and do not necessarily reflect the views of the funding agencies.

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

## A  TRAINING PROCEDURE

We list the training procedure of our method in Algorithm 1.

---

**Algorithm 1** Our proposed encoding-aware supernet training strategy

---

1: **Initialize.** Supernet weights $\mathbf{W}$, subnet weights $\mathbf{W}_i$, search space $\mathcal{A}$, subspace $\mathcal{A}_i$, the split iteration $T_s$, the max iteration $T_m$, a dataset $\mathcal{D}$.
2: **for** $t = 1 : T_s$ **do**
3:     Sample architecture and encoding strategies $a \in \mathcal{A}$.
4:     Sample a batch of graph data $\mathcal{D}_s \subset \mathcal{D}$.
5:     Calculate the training loss $\mathcal{L}_{train}$ over the sampled data.
6:     Update the supernet weights $\mathbf{W}$ through gradient descents.
7: **end for**
8: **for** $i = 1 : n$ **do**
9:     Let subnet $\mathbf{W}_i$ inherit the weights from supernet $\mathbf{W}$.
10:     **for** $t = T_s : T_m$ **do**
11:         Sample architectures and encoding strategies from the subspace $a \in \mathcal{A}_i$.
12:         Sample a batch of graph data $\mathcal{D}_s \in \mathcal{D}$.
13:         Calculate the training loss $\mathcal{L}_{train}$ over the sampled batch.
14:         Update the subnet weights $\mathbf{W}_i$ through gradient descents.
15:     **end for**
16: **end for**
17: **Output.** Subnets with weights $\mathbf{W}_i$.

---

## B  DATASET

We provide the statistics of the adopted datasets in Table 6 and Table 7.

Table 6: Statistics of graph classification datasets (precision) used to compare AutoGT with baselines. We adopt five datasets with relatively small numbers of graphs (upper part) and one dataset with a larger size (lower part) to demonstrate the efficiency of the proposed AutoGT.

| Dataset | #Graph | #Class | #Avg. Nodes | #Avg. Edges | # Node Feature | # Edge Feature |
|---------|--------|--------|-------------|-------------|----------------|----------------|
| COX2_MD | 303 | 2 | 26.28 | 335.12 | 7 | 5 |
| BZR_MD | 306 | 2 | 21.3 | 225.06 | 8 | 5 |
| PTC_FM | 349 | 2 | 14.11 | 14.48 | 18 | 4 |
| DHFR_MD | 393 | 2 | 23.87 | 283.01 | 7 | 5 |
| PROTEINS | 1,133 | 2 | 39.06 | 72.82 | 3 | 0 |
| DBLP | 19,456 | 2 | 10.48 | 19.65 | 41,325 | 3 |

Table 7: Statistics of graph classification datasets (AUC) used to compare AutoGT with baselines. We adopt two datasets with relatively small numbers of graphs (upper part) and one dataset with a larger size (lower part) to demonstrate the efficiency of the proposed AutoGT.

| Dataset | #Graph | #Class | #Avg. Nodes | #Avg. Edges | # Node Feature | # Edge Feature |
|---------|--------|--------|-------------|-------------|----------------|----------------|
| OGBG-MolBACE | 1,513 | 2 | 25.51 | 27.47 | 9 | 3 |
| OGBG-MolBBBP | 2,039 | 2 | 34.09 | 36.86 | 9 | 3 |
| OGBG-MolHIV | 41,127 | 2 | 24.06 | 25.95 | 9 | 3 |

## C  ADDITIONAL EXPERIMENTS

In Table 3, the results on COX2_MD and BZR_MD show larger standard deviations than other datasets. One plausible reason is that the number of graphs of these two datasets are relatively small, so that the results of the model can be sensitive to dataset splits. To obtain more convincing results on these two datasets, we conduct additional experiments by still utilizing 10-fold cross-validation for all

Table 8: Comparisons of AutoGT against state-of-the-art hand-crafted baselines. We report the average accuracy (%) and the standard deviation on all the datasets.

| Dataset | COX2_MD | BZR_MD |
|---|---|---|
| GIN | $57.22_{9.74}$ | $62.64_{8.23}$ |
| DGCNN | $60.33_{7.56}$ | $64.91_{9.42}$ |
| DiffPool | $59.52_{8.20}$ | $64.84_{8.51}$ |
| GraphSAGE | $53.62_{6.95}$ | $55.83_{8.27}$ |
| Graphormer | $59.22_{7.04}$ | $64.53_{9.43}$ |
| AutoGT(ours) | $\mathbf{63.45}_{8.04}$ | $\mathbf{67.18}_{9.87}$ |

Table 9: Comparisons of AutoGT with the different number of supernets. We report the average accuracy (%) and the standard deviation on the datasets.

| Dataset | PROTEINS |
|---|---|
| 1 supernet | $75.92_{3.10}$ |
| 2 supernet | $76.73_{3.25}$ |
| 4 supernet | $76.91_{3.35}$ |
| 8 supernet | $77.17_{3.40}$ |
| 16 supernet | $77.27_{3.65}$ |

the baselines and our proposed method, and repeat the 10-fold cross-validation with 10 random seeds. We report the results in Table 8. The results are consistent with Table 3, i.e., our method consistently outperforms other baselines, while the standard deviations are considerably smaller by adopting more repeated experiments.

In addition, to further explore how the number of supernets affects our proposed method, we carry out experiments with 1, 2, 4, 8, 16 supernets on the PROTEINS dataset, and report our results in Table 9. We can observe that as the number of subnets increases, the performance of our method increases. One possible reason is that more well-trained subnets can bring more consistent performance estimation results, which improves performance.

