# OpenReview forum: "AutoGT: Automated Graph Transformer Architecture Search"
_ICLR.cc/2023/Conference — ICLR 2023 notable top 5%_

### Official Review · Reviewer_iKrv · 2022-10-24

**Confidence:** 5
**Correctness:** 4
**Technical Novelty And Significance:** 3
**Empirical Novelty And Significance:** 3
**Recommendation:** 8

**Clarity, Quality, Novelty And Reproducibility:**

The paper is clearly written, with good quality and novelty. Reproducibility will be guaranteed if the authors promise to release the code.


**Details Of Ethics Concerns:**

None.

**Strength And Weaknesses:**

Strengths:
1. The novel idea of automating the design procedure of graph transformer.
2. Utilizing the neural architecture search for graph Transformers tends to become popular and practically important in graph representation learning community.
3. The proposed model is solid and the solution is appropriate for solving the challenges.
4. The paper is well written and the related work eases the understanding of neural architecture search and graph Transformer literature.

Weaknesses:
1. Fig. 1 is presented without explanation. Detailed explanations are expected to be provided.
2. For the architecture search and training step, the authors are encouraged to further discuss the difference from the proposed main ideas from the previous works on neural architecture search.


**Summary Of The Paper:**

This paper is the first work on automating the architecture design of graph Transformer, which is rarely studied in graph Transformer research. The contributions lie in solving the two challenges specifically related to graph-structured data during the design of neural architecture search.

**Summary Of The Review:**

This work proposes a neural architecture search based framework, Automated Graph Transformer (AutoGT), for automating the architecture design of graph Transformers. Tailored for non-euclidean graph data, the search space of AutoGT takes both the graph encoding strategy and Transformer architecture into considerations. Furthermore, the performance estimation strategy of AutoGT is designed to be encoding-aware through joint optimization of the encoding strategy and graph Transformer architectures. Experiments over several graph datasets with different sizes demonstrate the effectiveness of the proposed AutoGT. I think this paper investigates an interesting problem of good importance and I recommend to accept.

---

> ### Author Response · Authors · 2022-11-16
> **Response to Reviewer iKrv**
>
> We sincerely thank the reviewer for the positive feedback. We respond to the reviewer’s comments as follows.
>
> * **Q1. Figure 1 is presented without explanation.**
>
> **R1:** Thank you for your comment. We have added the explanation in Section 3.1 as follows: “The unified graph Transformer search space. The unified graph Transformer search space. It consists of the Transformer architecture space and the graph specific encoding space. The Transformer architecture search space is detailed in Table 1 The graph specific encoding search space is to decide whether each encoding strategy should be adopted or not and the threshold for the attention mask.” Please refer to Section 3.1 on page 3 of the revised version for details.
>
> * **Q2. For the architecture search and training step, the authors are encouraged to further discuss the difference from the proposed main ideas from the previous works on neural architecture search.**
>
> **R2:** Thank you for your suggestion. We would like to clarify the difference between the proposed method and previous works as follows. To the best of our knowledge, we are the first to study automating the designing of graph Transformer architecture. During the architecture training step, besides adopting the single path one-shot training strategy to reduce the computation cost as in [1], we also propose the supernet splitting strategy in the training process to deal with the coupling relations between Transformer architectures and the graph encoding. As a result, our proposed method can provide more consistent and fine-grained performance estimations. In comparison, previous neural architecture search works on non-graph Transformers such as [2, 3, 4] only use one supernet. As for the architecture search step, different from previous works on Transformer [2, 3, 4] which only consider the Transformer architecture, our proposed method considers both Transformer architecture and graph encoding during the evolutionary search.
>
> [1] Zichao Guo et al., Single path one-shot neural architecture search with uniform sampling. ECCV, 2020.
>
> [2] So, David et al., The evolved transformer. ICML, 2019.
>
> [3] Minghao Chen et al., Autoformer: Searching transformers for visual recognition. CVPR, 2021.
>
> [4] Chengyue Gong et al., NASViT: Neural Architecture Search for Efficient Vision Transformers with Gradient Conflict aware Supernet Training. ICLR, 2021.

---

### Official Review · Reviewer_ueHs · 2022-10-25

**Confidence:** 5
**Correctness:** 4
**Technical Novelty And Significance:** 3
**Empirical Novelty And Significance:** 3
**Recommendation:** 8

**Clarity, Quality, Novelty And Reproducibility:**

This paper is well written with comprehensive experimental evaluations over several well-used datasets. The idea proposed in this paper is novel. The paper is overall of good quality.

**Strength And Weaknesses:**

### Strengths:
+ The proposal of automatically searching the optimal graph transformer architecture is important in the community of machine learning.
+ Constructing a unified graph Transformer formulation by combining several typical transformer architectures and graph encoding strategies seems to be a novel contribution.
+ Taking graph-specific properties into the automated graph transformer architecture search process is helpful and can distinguish this paper from other approaches such as autoformer.
+ Experiments are conducted over several widely used and benchmark datasets.

### Weaknesses:
- The major focus in this work is the complex relations between transformer architectures and the graph encodings in transformer layer. The authors are suggested to further discuss why this problem is important and challenging, and how the authors address the problem specifically.


**Summary Of The Paper:**

This paper investigates the interesting problem of automated graph transformer through neural architecture search. The authors first propose a unified graph Transformer formulation that can represent most of the state-of-the-art graph transformer architectures. Then, an encoding-aware architecture searching and supernet training strategy are proposed. Experimental results over several graph datasets including the large-scale OGB datasets show the effectiveness of the proposed method.

**Summary Of The Review:**

This paper studies an interesting problem of automating the neural architecture design of graph transformer. The authors raise the challenges in the investigated problem, followed by the proposal of technical solid solution. Extensive experimental results validate the effectiveness of the proposed solution. I think this paper tends to be interesting to the research community and recommend acceptance for this paper.

---

> ### Author Response · Authors · 2022-11-16
> **Response to Reviewer ueHs**
>
> We sincerely thank the reviewer for the positive feedback. We respond to the reviewer’s suggestion as follows.
>
> * **Q1.  The authors are suggested to further discuss why this problem is important and challenging, and how the authors address the problem specifically.**
>
> **R1:** Thank you for your suggestion. We would like to further clarify the importance and challenges of this work. The main focus of this paper is to automatically design effective graph transformer architectures. This problem is important because graph transformers have shown outstanding performance in various graph tasks and our method can further enable the automatic designing of more effective graph transformers through neural architecture search, which is not explored in the literature. Meanwhile, this problem also brings the following challenges: i) the design of a unified search space for graph Transformers, ii) the coupling relations between Transformer architectures and the graph encodings. To address the problem, we first propose a unified graph Transformer search space, then propose a novel encoding-aware performance estimation strategy. Experimental results on various datasets also demonstrate the effectiveness of the proposed method.

---

### Official Review · Reviewer_C87i · 2022-11-02

**Confidence:** 3
**Correctness:** 4
**Technical Novelty And Significance:** 3
**Empirical Novelty And Significance:** 4
**Recommendation:** 8

**Clarity, Quality, Novelty And Reproducibility:**

Consistent quality

Table 3 tries to show the importance of the proposed evolution search by comparing it to simple random selection and existing hand-crafted options. Excellent experiment design! However, given the large deviation from the results, how can we ensure the proposed method results in consistently high-quality architecture designs? Without such a property, engineers will have concerns about deploying such a model to real-world applications.


Using eight subnets

The ablation study shows the performance boost from splitting the supernet into eight subnets. I would like to know how this behavior scales. Can you have better/worse results by having 2,4,16,32,64 subnets? You don't have to run all these setups, and some intuitive explanation serves my curiosity.


How to tune AutoGT

Since the proposed method aims to automatically provide graph transformer designs, AutoGT also has some hyper-parameters, for example, the number of supernet size and the search space. Can you justify why this makes the design process easier? How can you make sure the search space includes the right amount of options, not too many, not too few?


**Strength And Weaknesses:**

Strength
1. Well-explained basic transformer and graph encoding strategy, which show excellent intuition of the search space design.
2. The proposed framework is compatible with more existing graph encoding strategies, which allows a broader comparison between graph transformer designs.
3. Tab5 shows an ablation study using a single supernet vs. additional supernets. The ablation study shows the improvement of performance by splitting a supernet. Such a design brings extra performance without extra parameters.

Weakness:
1. Section 3 explained the proposed AutoGT well in text. However, neither Fig 1 nor Fig 2 are self-explanatory. I could not understand their relations with the proposed method without reading the main paragraphs. The small fonts in the figures made it more challenging.
2. Figure 2 was never mentioned in the text, which makes it tricky to understand which section it belongs to.
3. Table 3 tries to show the importance of the proposed evolution search by comparing it to simple random selection and existing hand-crafted options. The results have large deviations. How can we ensure the proposed method results in consistently high-quality architecture designs?
4. The proposed method aims to automatically provide graph transformer designs. However, AutoGT introduces extra hyper-parameters such as the number of supernet sizes and the search space's design.


**Summary Of The Paper:**

Existing graph transformers have succeeded in different applications but require cumbersome architecture designs and tuning by experienced engineers. This paper proposed a novel search framework for graph transformers, automatically searching within the designed unified search space for optimal architecture designs. The proposed method is compatible with most of the existing state-of-art graph transformers designs without introducing a substantial amount of computation cost. The experimental results show proposed strategy results in better designs over hand-crafted baselines.

**Summary Of The Review:**

This paper proposed AutoGT, which can automatically search the design space, which is compatible with most of the existing state-of-art graph transformers designs, without introducing a substantial amount of computation cost. The experimental results show proposed strategy results in better designs over hand-crafted baselines. However, I have concerns about the ease of use since AutoGT introduces extra hyperparameters that need to be tuned. And the auto designs from AutoGT have less consistent performance.

I recommend 6-marginally above the acceptance threshold.


----- After hearing back from authors 11/16 -----

Thanks for the explanation and new results. They cleared all of my concerns. Changing to "8: accept, good paper "

---

> ### Author Response · Authors · 2022-11-16
> **Response to Reviewer C87i (Part 1/3)**
>
> We sincerely thank the reviewer for the positive and detailed feedback. We have addressed all the comments. Please kindly find the detailed responses below.
>
> * **Q1. Figure 1 and Figure 2 are not self-explanatory, fonts in the figures are small.**
>
> **R1:** Thank you for your comment. To better clarify the figures, we have added additional explanations in the figure captions. Specifically, we have revised the caption of Figure 1 as follows: “The unified graph Transformer search space. It consists of the Transformer architecture space and the graph specific encoding space. The Transformer architecture search space is detailed in Table 1. The graph specific encoding search space is to decide whether each encoding strategy should be adopted or not and the threshold for the attention mask”. Besides, we have revised the explanation for Figure 2 as follows: “The framework of our work. Firstly, we construct the searching space for each layer, consisting of the Transformer architecture space (above) and the graph encoding strategy space (below). Then, we carry out our encoding-aware supernet training method in two stages: before splitting, we train a supernet by randomly sampling architectures from the search space, while after splitting, we train multiple subnets (inheriting the weights from the supernet) by randomly sampling architectures with fixed attention map augmentation strategies (except for the attention mask). Finally, we conduct an evolutionary search based on the subnets, and obtain our final architecture and results.” Lastly, we have enlarged the fonts in Figure 1 and Figure 2. Please refer to the revised version for details.
>
> * **Q2. Figure 2 was never mentioned in the text.**
>
> **R2:** Thank you for your comment. We have added the reference to Figure 2 at the beginning of Section 3 in the revised version. Please see page 3 in the revised paper for more details.

---

> > ### Author Response · Authors · 2022-11-16
> > **Response to Reviewer C87i (Part 2/3)**
> >
> > * **Q3. The results in Table 3 have large deviations.**
> >
> > **R3:** Thank you for your comment. We think one plausible reason that the standard deviations on datasets COX2_MD and BZR_MD are large is that the number of graphs of these two datasets is relatively small, so the results of the model can be sensitive to dataset splits [1]. Following your suggestion, to obtain more convincing results on these two datasets, we conduct additional experiments in Appendix C. Specifically, we still utilize 10-fold cross-validation for all the baselines and our proposed method and repeat the 10-fold cross-validation with 10 random seeds. We report the results as follows.
> >
> > | Dataset | COX2_MD | BZR_MD |
> > |  ----  | ----  | ----  |
> > | GIN | 57.22 ± 9.74 | 62.64 ± 8.23 |
> > | DGCNN | 60.33 ± 7.56 | 64.91 ± 9.42 |
> > | DiffPool | 59.52 ± 8.20 | 64.84 ± 8.51 |
> > | GraphSage | 53.62 ± 6.95 | 55.83 ± 8.27 |
> > | Graphormer | 59.22 ± 7.04 | 64.53 ± 9.43 |
> > | AutoGT | 63.447 ± 8.04 | 67.18 ± 9.87 |
> >
> > The results show that the standard deviations are considerably smaller by adopting more repeated experiments. We have added the experiments and corresponding analyses in Appendix C of the revised paper.
> >
> > [1] Damien Brain et al., On the effect of dataset size on bias and variance in classification learning.
> >
> > * **Q4. AutoGT aims to automatically provide graph transformer designs, but it  introduces extra hyper-parameters.**
> >
> > **R4:** Thank you for your comment. We agree that designing a proper search space sometimes will inevitably introduce extra hyper-parameters, which is a common issue for neural architecture search. Nevertheless, in our proposed method, we mainly introduce two sets of extra hyper-parameters: the number of supernets and hyper-parameters in the design of the search space. The former is to improve the proposed method’s performance by dealing with the coupling relations between Transformer architectures and the graph encoding while the latter is the common procedure of conducting neural architecture search. That is to say, our method doesn’t introduce too many hyper-parameters, which is on par with the graph Transformer without automated design. On the other hand, our method is not very sensitive to hyperparameters. Extensive studies in Appendix C show that our method can consistently outperform other state-of-the-art baselines by a large range with different choices of the number of supernets.

---

> > > ### Author Response · Authors · 2022-11-16
> > > **Response to Reviewer C87i (Part 3/3)**
> > >
> > > * **Q5. Will the results be better or worse by having 2,4,16,32,64 subnets?**
> > >
> > > **R5:** Thank you for your comment. Following your suggestions, we have further conducted experiments on having 2, 4, and 16 subnets in addition to the original experiments. The experimental results for the PROTEINS dataset are shown as follows.
> > >
> > > |  # Subnet | 1 | 2 | 4 | 8 | 16 |
> > > | ---- | ---- | ---- | ---- | ---- | ---- |
> > > | PROTEINS | 75.92 ± 3.10 | 76.73 ± 3.25 | 76.91 ± 3.36 | 77.17 ± 3.40 | 77.27 ± 3.66 |
> > >
> > > We can observe that as the number of subnets increases, the performance of our method increases. One possible reason is that more well-trained subnets can bring more consistent performance estimation results, which improves performance. We have added the experimental results and analyses in Appendix C in the revised paper.
> > >
> > > * **Q6. How to make sure the search space includes the right amount of options?**
> > >
> > > **R6:** Thank you for your question. We agree that enabling neural architecture search methods to guarantee obtaining architectures with the right amount of information is a challenging and open problem in AutoML. Nevertheless, our proposed method can make finding high-performance architectures easier, as validated by our experimental results that our method greatly outperforms hand-designed methods. Besides, we would like to clarify that we only adopt two sets of hyper-parameters, one for relatively small-scale graphs with a smaller number of possible architectures (AutoGT_base with 7e18 architectures) and the other for relatively large-scale graphs with more possible architectures (AutoGT with 1e39 architectures). Our design principle is that the number of architecture parameters should be consistent with the number of graphs, and our experimental results show that these two sets of hyper-parameters can work for various graph datasets as well.

---

### Decision · Program_Chairs · 2023-01-20

**Decision:**

Accept: notable-top-5%

**Justification For Why Not Higher Score:**

N/A

**Justification For Why Not Lower Score:**

This paper receives three very positive ratings (8 8 8) and is among top 2.7% according to the conference reviewing statistics.
More importantly, this paper studies a unique and interesting research problem, automated graph transformer architecture search and propose a non-trivial and effective method for doing it.

**Metareview: Summary, Strengths And Weaknesses:**

This paper introduces a new research problem, Automated Graph Transformer Architecture Search, and proposes a unified graph transformer formulation for existing graph transformer architectures. Then a novel architecture search method that can jointly optimize Transformer architecture and graph encoding strategies is proposed. The problem and the method are well motivated and formulated, witch extensive experimental supports. The paper is also well written. All the (minor) concerns are addressed after the rebuttal, and the reviewers unanimously suggest acceptance for this work. The AC also believes this work studies an important problem and is a solid contribution to the ICLR community.

**Note From Pc:**

if the above contains the word "oral" or "spotlight" please see: "oral" presentation means -> notable-top-5% and "spotlight" means -> notable-top-25%. As stated in our emails, we are disassociating presentation type from AC recommendations